# The Associations between Evacuation Movements and Children’s Physiological Demands Analyzed via Wearable-Based Sensors

**DOI:** 10.3390/s22218094

**Published:** 2022-10-22

**Authors:** Bo Zhang, Xiaoyu Gao, Jiaxu Zhou, Xiaohu Jia

**Affiliations:** 1Architecture College, Inner Mongolia University of Technology (IMUT), Hohhot 010051, China; 2UCL Institute for Environmental Design and Engineering, The Bartlett, University College London (UCL), London WC1H 0NN, UK; 3Inner Mongolia Key Laboratory of Green Building, Architecture College, Inner Mongolia University of Technology (IMUT), Hohhot 010051, China

**Keywords:** children movement assessment, evacuation postures, physiological demand, wearable sensors

## Abstract

During fire evacuations, crawling is recommended to prevent harm from toxic smoke and to access more breathable air. Few studies have evaluated the physiological burden of crawling, especially for children. The method of using wearable sensors to collect data (e.g., electrodermal activity, EDA; skin temperature, SKT) was used to evaluate the effects of different locomotive postures on children’s velocity and physiological demands. Twenty-eight (28) children (13 boys and 15 girls), aged 4 to 6 years old, traveled up to 22.0 m in different postures: Upright walking (UW), stoop walking (SW), knee and hand crawling (KHC). The results showed that: (1) Gender and age had significant impacts on children’s velocity (*p* < 0.05): Boys were always faster than girls in any of the three postures and the older the child, the faster the velocity for KHC. (2) Physiological results demonstrated that KHC was more physically demanding than bipedal walking, represented by higher scores of the EDA and SKT indicators, similar to the findings of adults. (3) Gender and age had significant impacts on children’s physiological demands (*p* < 0.05). The physiological demands were greater for boys than girls. In addition, the higher the age, the less physiological demands he/she needs. Overall, the findings suggest that children are unnecessarily required to choose crawling precisely as adults as the best posture to respond to emergency scenarios. In a severe fire, stoop walking is suggested, as there is more respired air and children could move quickly and avoid overworking physiological burdens. The results of this study are expected to be considered in the evaluation of current evacuation recommendations and for the safety guide of preparedness to improve the effectiveness of risk reduction for children.

## 1. Introduction

In recent years, kindergarten fires have occurred frequently and caused a large number of casualties. Besides the tragic loss of life and impact on children’s families and communities, children’s injuries and fatalities have been shown to result in significant economic costs. During emergency situations (e.g., fires), smoke inhalation is the most common cause of death in fire situations [1,2]. As the smoke spreads from the room ceiling, the height of the smoke layer drops down and it is necessary to change locomotive postures to evacuate safely. Researchers have found that adopting proper locomotive postures under different scenarios is critical for improving the survival rate of adults [3]. However, research on children is sparse. Moreover, children’s physical skills are considerably different from adults which may result in a more profound effect in different evacuation postures [4].

Some studies suggested that crawling during fire events may result in cardiac overload and increase the occurrences of heart-related risk in adults [5,6]. However, few studies have examined the risks and physiological demands of the evacuation of children in different postures. According to the 2019 World Census, there are 1.97 billion children aged 0–14 years old worldwide, among which 220 million are in China [7]. Therefore, there is a strong need to study the capacity of children to take on different evacuation postures to reduce children’s risk in evacuation.

The purpose of this article is to assess the risk of different postures on children’s physiological demands during evacuation using wearable sensors. It was designed to assess the children’s risk in fire evacuation, quantify the hazard of this study as fire, then, make a randomized cross-over trial on other elements of the risk, namely exposure and vulnerability.

### 1.1. Evacuation Postures

Previous studies were mostly conducted on adults with an emphasis on bipedal walking. Only a few studies have analyzed crawling behavior, all of which were conducted in adults [8,9,10,11,12,13,14,15]. As shown by these studies, crawling is significantly slower than walking. In addition, the crawling velocity may be affected by gender and body conditions such that overweight individuals and females generally have a lower crawling velocity [8,9]. Muhdi et al. (2006) showed that the average and maximum velocities of knee and hand crawling are 0.71 m/s and 1.47 m/s, respectively [12]. Nagai et al. (2006) reported that adults have an average crawling velocity of 0.73 m/s and a significantly higher upright walking velocity of 1.2 m/s [13]. Wang et al. (2020) analyzed the velocity of middle school and primary school students in crawling. They found that middle school and primary school students had a higher crawling velocity than undergraduate students [16]. Most studies have only considered upright walking when analyzing the evacuation behavior of children [17,18,19,20,21]. Furthermore, simulation-based studies of children’s evacuation behavior were also mostly conducted using the upright posture [22,23,24,25]. However, taking on different evacuation postures can significantly affect evacuation velocity, and there are limited studies on physical characteristics in different postures. Thus, it is necessary to use advanced equipment to evaluate behavioral characteristics of children with different evacuation postures. The data could provide an important basis to establish a dynamic evacuation model for children.

### 1.2. Physiological Demands

The physiological demands of bipedal exercise for adults (e.g., walking, jogging, or running) have been investigated in past studies. The results showed that walking imposes fewer physiological demands than other types of exercise [26,27,28,29,30,31,32]. Morrissey et al. (1985) analyzed the metabolic cost of stoop walking and crawling. Their study revealed an increased metabolic cost with an increasing level of waist bending during stoop walking [33]. For example, a higher heart rate and a stronger cardiorespiratory performance are required for firefighters during crawling [34]. It has also been shown that crawling imposes higher physiological demands and could result in physical discomfort (Moss, 1934) [35]. Gallagher et al. (2011) investigated the behavior of stoop walking, two-point crawling (knees only), and four-point crawling (knees and hands) in restricted spaces. Their study showed a significant difference in the kinematic performance and physiological demands between different postures [36]. Additionally, the heart rate (HR), oxygen consumption (VO2), and respiratory exchange ratios (RERs) of adults are substantially higher during crawling than bipedal exercise [11,37]. Merla et al. (2005) monitored the skin temperature of athletes and found that the skin temperature starts to drop at the beginning of exercise and drops by about 3–5 °C at the end of exercise [38]. However, we still do not know the relationship between the evacuation movement and children’s physiological demands. Therefore, it is crucial to perform a quantitative study on the physiological demands of children during evacuation using different postures. Such an analysis will provide important information to evaluate children’s risk of evacuation.

Recent International Building Code (IBC) (2018) standards require that the distance of kindergartens to an exit should not exceed 22.86 m if a sprinkler system is in place [39]. In China, similar evacuation distance standards for kindergartens have also been implemented. The evacuation door installed between two exits must be located within a straight-line distance of 25 m [40]. However, the physiological risk of children during evacuation has not been analyzed in past studies. Some emerging evidence suggested age and gender impact on evacuation performance in adults [11,41,42], however the role of gender and age difference in relation to children’s evacuation performance is not clear. In addition, with the continuous development of science and technology, it becomes necessary to investigate the relationship between evacuation postures and associated physiological demands using wearable sensors in order to reduce risks for children and improve evacuation performance.

### 1.3. Aims and Contributions

The purpose of this article was to assess the risk of different postures on children during evacuation by analyzing the evacuation velocity and physiological indicators in three different postures (upright walking, UW; stoop walking, SW; knee and hand crawling, KHC). In this study, an experimental analysis was performed to understand the physiological condition of children during evacuation. The physiological demands were quantitatively analyzed based on body signals (e.g., electrodermal activity and skin temperature) collected by wearable sensors. Specifically, the focus of this study was to answer the following questions:Does age/gender have an effect on children’s evacuation velocity?How are children’s physiological demands affected by different postures?What is the impact of age/gender on children’s physiological demands?

## 2. Materials and Methods

In this study, children’s physiological indicators were monitored by wearable sensors. Each subject completed the trial using three different postures randomly: (1) upright walking, UW; (2) stoop walking, SW; and (3) knee and hand crawling, KHC (Figure 1). The experiment was recorded by a camera for behavioral observation (Figure 2). The relationship between the physiological indicators and its confidence level were analyzed by IBM SPSS Statistical 25.0.

### 2.1. Experimental Setting and Equipment

The experiment was conducted in Hua Di Kindergarten (in Hohhot, China), as the kindergarten exhibits the typical features of general urban kindergartens based on preliminary analysis of Chinese preschools [43,44]. The experiments were conducted in the third-floor corridor where there is less external interference.

The corridor is 25 m in length and 1.8 m in width. The test track, with a total length of 22 m, was divided into six segments in order to detect the potential changes of evacuation velocity. The start and finish lines were set 1.5 m from the beginning and the end of the track to control any acceleration and deceleration effects (Figure 2) [45]. A camera mounted on the trolley was used to record the experiment. The physiological data (e.g., electrodermal activity, EDA; skin temperature, SKT) of the experiment subjects were processed in ErgoLAB software.

### 2.2. Participants

Tatuic and Dederichs (2013) revealed that the minimum age for preschool at which children can understand and can carry out simple instructions is 2.5–3 years old [46]. Therefore, the subjects of this study were 28 children aged 4–6 years who were selected from senior classes. The inclusion criteria were as follows. They were in good health and did not manifest any COVID-19 symptoms. None of the subjects took any psychotropic drugs and were free from any musculoskeletal injury or cardiovascular disease. The details of subjects are shown in Table 1 [45]. The experimental data were collected with the consent of Hua Di Kindergarten and the parents of children. All parents signed an informed consent form before the test. Ethical approval for this study was provided by the Ethics Committee of the College of Architecture, Inner Mongolia University of Technology.

### 2.3. Physiological Measurements

In this experiment, the EDA and SKT data of the children were collected as the main means of analyzing children’s physiological demands. The principle of EDA measurements has been discussed in detail in our prior publication [43]. The collection of SKT measurement was carried out as follows:

The SKT in the target area was measured directly by attaching a temperature sensor to the epidermis. The skin temperature was measured at the finger to prevent any undesired impact on body activity. During energy metabolism, most of the energy released from the human body is converted into heat, thereby regulating the body temperature. The human body temperature refers to the temperature inside the human body, which remains relatively constant. While the body temperature may be affected by the metabolism, it only varies slightly over time. On the other hand, skin temperature changes much more significantly than body temperature as the skin is in contact with the external environment directly. The skin temperature is affected by both the metabolism in the human body and the heat exchange between the body and its environment.

### 2.4. Experimental Procedure

Each experiment was conducted in the corridor with a single subject, while the other subjects waited in the classroom. Therefore, each subject was unaware of the upcoming activity before the experiment. The experimental timetable was from 10:00 am to 11:30 am and from 3:00 p.m. to 4:30 p.m. each day. Children were required to rest for at least 2 h before the experiment to prevent experiencing the effects of strenuous exercise or tiredness.

Before the experiment, the age, gender, height, and weight for every subject were recorded. The blue numbers were assigned for boys and the red numbers for girls. The sensors and the skin of the subjects were sterilized in advance to remove impurities that may have impeded the sensing function of the electrodes on the sensor. The wearable sensors (e.g., EDA and SKT) with a frequency of 64 HZ were attached to the appropriate part of their body with straps and calibrated carefully by researchers (Figure 3). After checking that all the sensors were installed in place, the researchers turned on the sensors to pair with the computer for signal reception. Then, the children were instructed to sit down in a chair to wait to begin. At the same time, the physiological data were collected by wearable sensors as baseline value [33,43]. Baseline measurements of each child’s physiological data were performed before each trial. The recording process of the physiological signals is shown in Figure 4.

During the experiment, each subject participated in three separate trials (up to 22.0 m each) using three different postures including: (1) upright walking (UW); (2) stoop-walking (SW); and (3) knee and hand crawling (KHC). The three different postures were assigned in randomized order to negate potential order effects (i.e., learning and fatigue). For the stoop walking posture, it is hard for children to maintain a consistent bending angle during the evacuation as children’s balance is not fully developed. We told the children before the experiment to maintain the stooping position as much as possible during the entire stoop walking trial. Children need to wear adjustable knee pads, elbow pads, and gloves to protect their safety during KHC. During the experiment, an investigator closely followed each subject while pushing a trolley with a digital video camera at a frame rate of 30 frames per second (FPS) and a computer mounted to record the experiment (Figure 5). The recorded video was used to determine the intermediate times and velocities as the subjects passed over each track section. The subjects were provided sufficient rest between successive trials to reduce the effects of tail retention and to avoid confounding effects caused by fatigue. The subjects could ask for rest or to stop the experiment at any time during each trial. Stopping criteria for the trials were: any physical discomfort experienced by the subject; malfunction of the test equipment; or a trial termination request by the subject [11].

### 2.5. Data Analysis

During the crawling, one girl asked for termination due to physical exhaustion during KHC, so the data of a total of 27 subjects were collected. A dataset containing the final measurement results was created using IBM SPSS 25.0 (IBM, New York, NY, USA) [9,29] to analyze the effects of different postures on children’s physiological demands and evacuation velocities.

The EDA indicators (skin conductivity, SC; skin conductance level, SCL; skin conductance response, SCR) and SKT indicators of the 27 subjects at the baseline, and during the experimental evacuation were recorded. Data analysis was performed by the following methods:

Based on 95% confidence level, the statistical significance of the difference between the scores was evaluated to determine the differences among the physiological data corresponding to different evacuation postures.

All data points were normalized relative to the baseline data of the resting state and all individual differences between subjects were eliminated so that the processed data could be compared with the results of the evacuation.

In this study, the distance for every subject was 22.0 m, and the evacuation velocity was calculated by the formula below:V=S/ΔT
where *V* (m/s) stands for the velocity, *S* (m) stands for distance, and ΔT (s) stands for the time.

## 3. Results

### 3.1. Evacuation Velocities of Children in Different Postures

#### 3.1.1. Velocity

Figure 6 shows the average evacuation velocities of all subjects in different postures.

ANOVA analysis indicates there is a significant difference in evacuation speed of children between the three posture conditions (*p* < 0.05). The average evacuation velocities for UW, SW, and KHC were 2.29 ± 0.53 m/s, 2.07 ± 0.62 m/s, and 0.58 ± 0.20 m/s, respectively [45]. The mean and SD values of velocity were analyzed together with ergonomic risk assessment of children in our previous work (Jia et al., 2021). We report this for clarity and transparency.

#### 3.1.2. Age

The ANOVA tests showed a significant difference between the age and evacuation velocity (*p* < 0.05) (Figure 7). The relationship of bipedal walking velocities between 5-year-old and 6-year-old was very close and was much faster than the velocity of 4-year-old children. However, in terms of KHC, the older the child, the faster was the velocity.

#### 3.1.3. Gender

According to the results of independent *t*-tests, Gender exhibited a significant effect on travel velocity (*p* < 0.05). As shown in Figure 8, the average velocities of boys were faster than girls, with boys being 21.67%, 22.10%, and 47.83% faster than girls in UW, SW, and KHC, respectively. It is worth noting that all girls yielded similar velocity in the KHC condition, but the KHC velocity for boys had a wider distribution (for the complete list, see Appendix A). One potential explanation might be due to the difference in body development, physical or biological energy consumption of the body between boys and girls.

### 3.2. The Significance of Influence of Evacuation Postures on Physiological Indicators

#### 3.2.1. Electrodermal Activity, EDA

Table 2 included the EDA indicators at baseline and during evacuation. Results demonstrated that evacuation behavior affected the EDA scores. The values of EDA for all three types of evacuation postures were higher than the baseline. Paired *t*-tests were conducted to verify the significant difference among the SC, SCL, and SCR of EDA indicators. According to the results of paired *t*-tests (Table 3), evacuation postures exhibited a significant effect on children’s SC and SCL (*p* < 0.05), which could be used as the main indicators of analyzing children’s physiological demands. The results also indicated that there was no significant difference in SCR between KHC and baseline (*p* > 0.05). Therefore, the SC and SCL were used to analyze the association between physiological demands and different types of evacuation postures (UW, SW, and KHC).

#### 3.2.2. Skin Temperature, SKT

Table 2 summarizes the average SKT for all subjects in three different postures. It can be seen that the SKT measured during evacuation was significantly lower than the baseline value, indicating that the evacuation behavior caused changes to children’s SKT. Paired *t*-tests were further performed to verify the level of significance (Table 3). The test results showed that the evacuation postures had a significant effect on the children’s SKT (*p* < 0.05). Therefore, the SKT indicator could be used to analyze whether different evacuation postures affect children’s physiological demands.

### 3.3. Effects of Different Evacuation Postures on Children’s Physiological Demands

According to the paired *t*-test results described above, the effects of different evacuation postures on children’s physiological demands were analyzed by EDA indicators (SC and SCL) and SKT indicators. The means of all physiological indicators at baseline condition and during the evacuation in three different postures were calculated (Table 2). The effects of different evacuation postures on each physiological indicator were determined based on the data presented in Figure 9 and Figure 10.

Figure 9 presents the changes in the mean values of the EDA indicators for different evacuation postures. In terms of SC, the value measured during the evacuation is significantly higher than the baseline value. In particular, knee and hand crawling (KHC) had the highest SC value, followed by stoop walking (SW) and upright walking (UW).

Compared to the baseline value, the SC value was increased by 94.15%, 97.95%, and 115.79% during evacuation in UW, SW, and KHC postures, respectively. Similarly, it can be seen from the SCL changes that the trends of SCL in different experimental scenarios were the same as SC. The KHC had the highest SCL value, followed by the SW and UW. Compared to the baseline value, the SCL was increased by 78.88%, 88.03%, and 105.72% during the evacuation in UW, SW, and KHC, respectively. Therefore, the evacuation postures can increase the values of the EDA indicator, which means that the evacuation behavior can promote secretion by the sweat glands to improve the skin conduction level. KHC had the most significant effect.

As shown in Figure 10, the SKT measured during the evacuation was significantly lower than the baseline value. Compared to the baseline value, the SKT was reduced by 3.28%, 7.79%, and 8.66% in UW, SW, and KHC, respectively. This trend suggested that the evacuation behavior could cause the skin temperature to decrease due to the evaporation of sweat on the skin surface. In particular, the effect of KHC was the most significant, followed by SW and UW.

### 3.4. Effects of Age/Gender on Children’s Physiological Demands

According to the results of the previous section, children in KHC were caused the greatest physiological demands during the evacuation compared to bipedal walking. Therefore, this section analyzes and discusses the effects of age and gender on children’s physiological demands based on the data obtained in KHC.

#### 3.4.1. Age

Here, the age difference (4–6 years old)) in relation to children’s physiological demands during the evacuation process is investigated. The scores of EDA and SKT indicators of all children in KHC were different by age. ANOVA results demonstrated that children’s physiological demands were affected by age, as the EDA and SKT indicators were significantly different among different ages (*p* < 0.05).

Table 4 summarizes the scores of EDA and SKT indicators and associated relative change at baseline and during evacuation in KHC based on age. The associated relative change was calculated by Formula (1).

Relative change (%) = [(KHC test value − baseline value)/baseline value] × 100
(1)


As shown in Table 4, the SC and SCL values of the EDA indicator during evacuation in KHC were significantly high than those at baseline. The relationship between age and the relative change of SC and SCL was the older the children, the less the relative change they underwent from the experiment. In terms of SC, the relative change was 6.20% higher for four-year-olds than for five-year-olds, and 21.53% higher for five-year-olds than for six-year-olds. The relative change of SCL was 18.41% greater for 4-year-old children than for 5-year-old children, and 22.22% greater for 5-year-old children than for 6-year-old children. Similarly, the values of the SKT indicator of children during evacuation in KHC were significantly lower than those at baseline. The order of relative change of SKT was as follows: 4-year-olds > 5-year-olds > 6-year-olds. From the perspective of SC and SCL of EDA indicators and SKT indicators, these results demonstrated that as children age, they have less physiological demands on them during evacuations in KHC.

#### 3.4.2. Gender

For children aged 4–6, does gender affect the children’s physiological demands in KHC? To explore this issue, all subjects in KHC were gender-differentiated, while independent *t*-tests at a 95% confidence level were performed.

Table 5 summarizes the physiological indicators of the subjects at the baseline condition and during the evacuation. Among EDA indicators, the relative change of scores in boys was higher than in girls, with a relative change of SC and SCL in boys being 6.25% and 6.62% higher than in girls, respectively. The SKT indicator of boys changed more substantially than girls, with the relative change in SKT scores of boys being 8.04% higher than girls.

The gender-disparity analysis showed that the physiological indicators (EDA and SKT) were significantly affected by gender (*p* < 0.05), and boys had greater physiological demands on them in KHC than girls.

## 4. Discussion and Limitation

This study provided novel venues with wearable technology for a better understanding of children’s evacuation behavior and may have an impact on children’s health and safety. Based on an on-site experimental analysis of the three postures, this study analyzed the evacuation velocities and physiological demands of 28 children aged 4–6 years, by showing differences in velocity and physiological indicators (e.g., EDA and SKT) between different evacuation postures and the effects of age and gender.

As shown by the results, the evacuation velocity was significantly affected by evacuation postures, age, and gender. The velocity of KHC was significantly slower than bipedal walking velocities. Similar results could be found in previous children’s studies [8,9]. A recent study conducted by Cao et al. (2018) suggested the mean velocity of UW, SW, and KHC for adults was 1.93, 1.84, and 0.84 (m/s), respectively [11]. For comparison, our study found that the evacuation velocities of children were faster than those of adults in UW (2.29 m/s) and SW (2.08 m/s), however, the KHC velocity (0.58 m/s) of children was slower than that of adults. This study focused on individual behavior rather than group behavior, to avoid collisions. According to the velocity quadratic fitting, the evacuation velocity started to decline after traveling up to 11.60 m in KHC. The result was different from the previous studies on adults [11]. According to Muhdi et al. (2006), the distance of adults at which speed starts to fall is 2.46 m shorter than that of children [12]. The older the child, the faster the velocity, but there were no relevant findings for adults. The velocities for boys were faster than girls, which is the same as the results of adult-related studies [8,9,11].

Based on the results presented above, there was a significant difference between children’s physiological demands and associated physiological indicators (e.g., EDA and SKT). The discovery was similar to previous research on the association between different types of alarm sounds and children’s physiological responses [43]. The new finding is that the physiological effects of different evacuation postures on children are much greater than those of alarm sounds. One potential explanation for the difference is that a larger number of muscles and energy are required for physical movements than for sound perception.

In previous studies, only children’s walking velocity was measured [8,12]. They did not consider the physiological effects of different movement postures on children. In this experiment, we found that EDA, SKT, and velocity were significantly affected by different movement postures. Crawling was perceived to be more difficult than walking. This was consistent with the results of a previous study on 24 college students by Li et al. (2018) [11]. We found that the relationship of physiological demands between crawling and walking could also be seen in children (aged 4–6 years old). In addition, the finding of physiological demands of boys greater than girls during the evacuation was consistent with the results of adults [10]. The younger the evacuee, the greater was the change in the physiological demands during evacuation and the greater the risk. In other words, younger evacuees have greater physiological demands and potential risks than older ones, but the relevant findings were not found for adults. In addition, boys have greater physiological demands and take more potential risks during evacuation than girls. In the event of a fire disaster, the smoke will not only affect lung capacity and block eyesight but also reduce the evacuee’s evacuation velocity [47,48,49]. Smoke will increase the physiological demands on the evacuees, which increases the challenge of the evacuation. Therefore, when analyzing and designing evacuation paths for children in future studies, the effects of different evacuation postures and the corresponding physiological features should be taken into consideration to reduce the evacuation risks for children.

The major contribution of this study was to obtain children’s evacuation velocity and physiological demands under different evacuation postures by statistically analyzing the experimental data. Furthermore, this study demonstrated the feasibility of continuously monitoring the impact of different evacuation postures on children’s physiological demands using multi-channel wearable sensors. Considering the complex and dynamic requirements of children during evacuations, monitoring their physiological costs by an objective, continuous, and non-invasive method will help us better understand children’s evacuation behavior. This study of using EDA and SKT indicators as observational variables to record the physiological condition of the subjects provides a new way of evaluating the fatigue state of the human body and has a promising application when combined with wearable devices.

Thus, it is hard to train children exactly like adults for fire evacuation. In both aspects, evacuation velocity and physiological demands, stoop walking is better than knee and hand crawling. Compared to upright walking, children could obtain more respirable and clean air as smoke fills the space from the ceiling downwards. Therefore, updating the response criterion, the knowledge, and the safety guide of preparedness is necessary to improve the effectiveness of children’s risk reduction during a fire. There is a need to formulate scientific emergency planning on firefighting and emergency evacuation and train children to pick a better posture based on their physical condition, such as their age, gender, and physiological demands during the fire evacuation when they are doing fire drills. Thus, children will respond effectively during the actual evacuation after the training and reduce children’s risk during the evacuation firsthand.

There are some limitations that should also be acknowledged in this study. First, the subjects were selected from children aged 4 to 6 years, while children older than 6 years were not considered. Second, we only analyzed the EDA and SKT indicators but did not consider other physiological factors (e.g., rates of oxygen consumption, respiratory exchange ratios). Third, the material of surface for each trial was the same in this experiment. Future studies considering the effect of different materials and slopes of surface on children’s velocity might become necessary. Moreover, there were only 28 children involved in this study and the height and BMI of the subjects did not vary significantly. Future studies which include a large sample size in response to consideration of the physiological and evacuation performance.

## 5. Conclusions

To minimize the children’s potential risk in a fire event, a scientific evacuation posture should be used for reducing the risk of exposure and vulnerability. In this study, the effects of the evacuation postures on children’s evacuation velocity and physiological demands were evaluated by analyzing the velocity, EDA and SKT indicators collected from 28 children (13 boys and 15 girls) aged 4–6 years old through wearable sensors. The following conclusions can be drawn from the data analysis:(1)Age (4–6 years old) and gender had significant effects on children’s evacuation velocities; the older the children, the faster the velocity in crawling. In terms of gender, boys were always faster than girls.(2)Crawling is more physically demanding than bipedal walking for children, evidenced by the higher relative change in the scores of the EDA and SKT indicators.(3)Age (4–6 years old) and gender had significant effects on children’s physical demands; the older the child, the less physiological demands he/she needs. Furthermore, boys had greater physiological demands than girls.

## Figures and Tables

**Figure 1 sensors-22-08094-f001:**
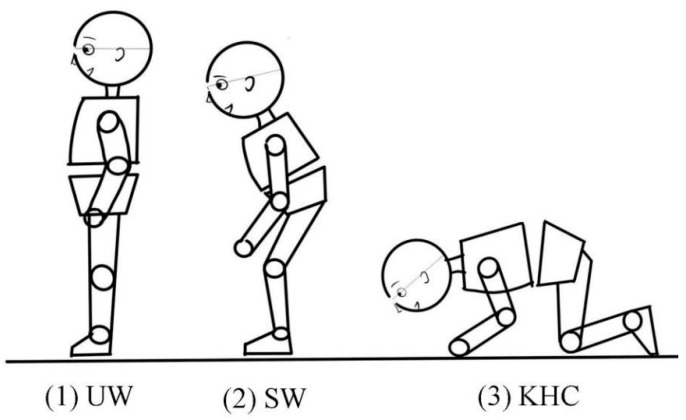
Evacuation postures. (1) UW: upright walking; (2) SW: stoop walking; (3) KHC: knee and hand crawling.

**Figure 2 sensors-22-08094-f002:**
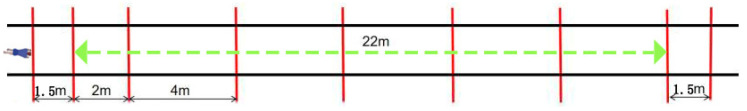
Test track.

**Figure 3 sensors-22-08094-f003:**
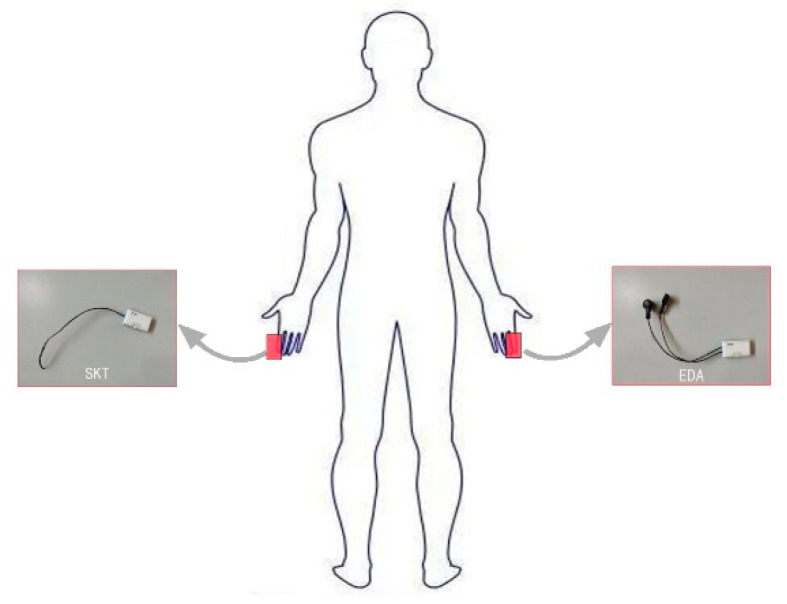
Wearable sensor placement sites.

**Figure 4 sensors-22-08094-f004:**
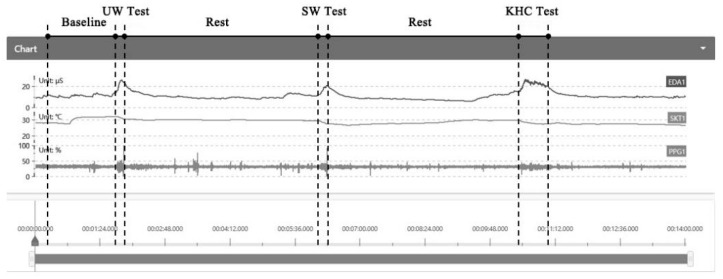
Interface of the experimental data collection platform.

**Figure 5 sensors-22-08094-f005:**
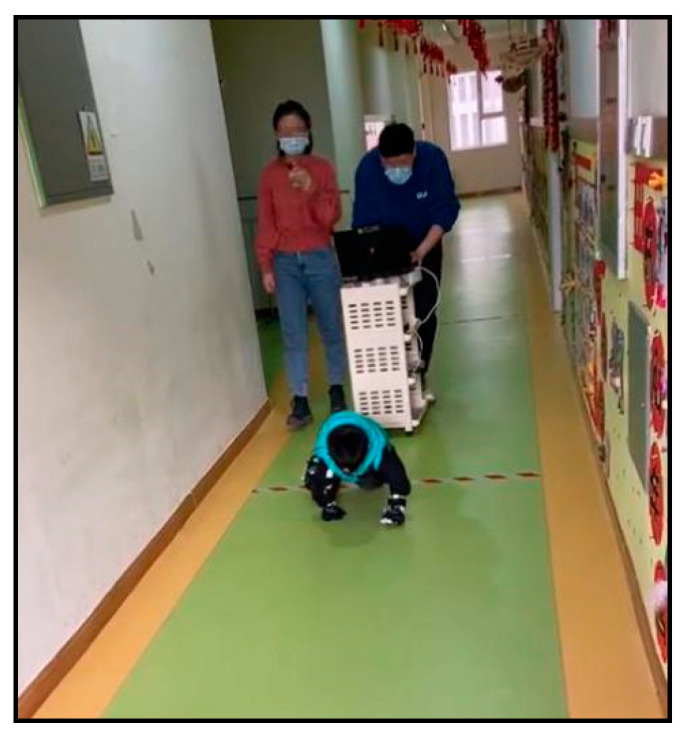
Digital camera used to record the entire trials.

**Figure 6 sensors-22-08094-f006:**
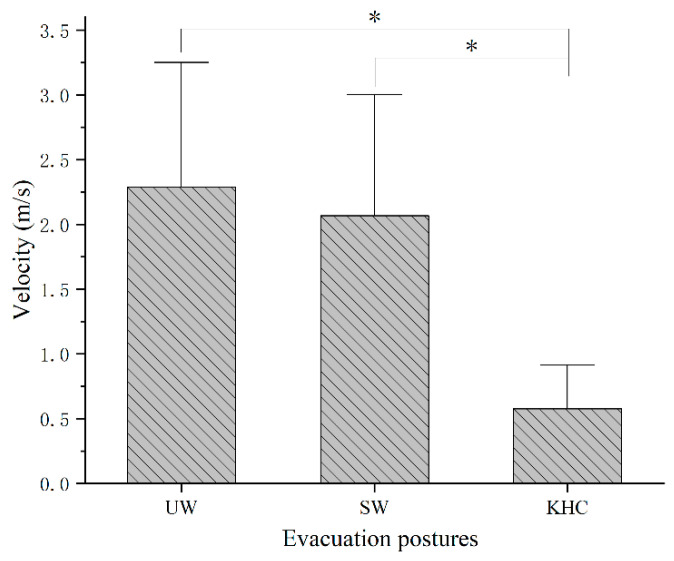
Average evacuation velocities for different postures. UW—upright walking; SW—stoop walking; KHC—knee and hand crawling. (* denotes significant difference between evacuation postures).

**Figure 7 sensors-22-08094-f007:**
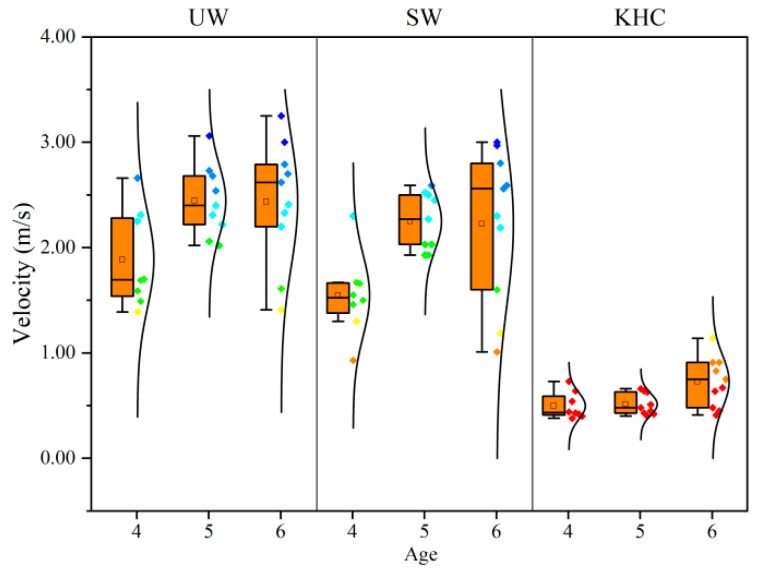
Average evacuation velocities for different ages. UW—upright walking; SW—stoop walking; KHC—knee and hand crawling. The number of subjects for four years old, five years old, and six years old was 8, 9, and 10, respectively.

**Figure 8 sensors-22-08094-f008:**
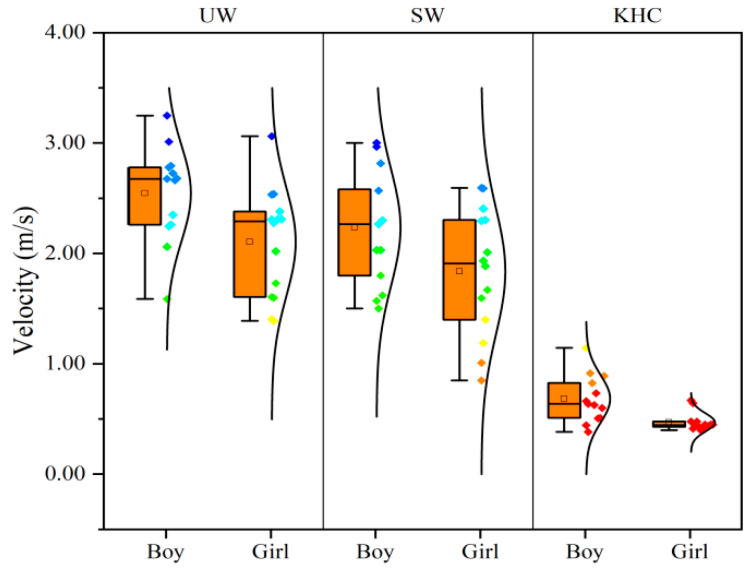
Average evacuation velocities for different gender. UW—upright walking; SW—stoop walking; KHC—knee and hand crawling. The number of subjects for boys and girls was 13 and 14 respectively.

**Figure 9 sensors-22-08094-f009:**
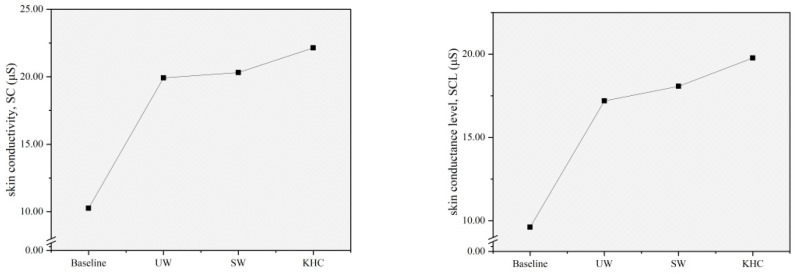
Changes in the mean value of EDA indicators by postures. UW—upright walking; SW—stoop walking; KHC—knee and hand crawling. SC: Skin conductivity; SCL: Skin conductance level.

**Figure 10 sensors-22-08094-f010:**
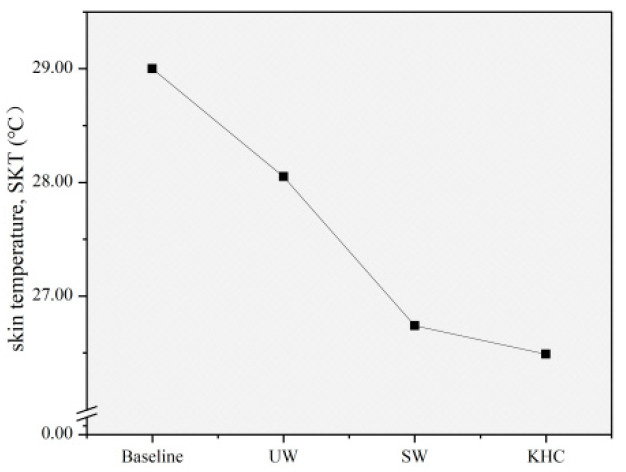
Changes in the mean value of SKT indicators by postures. UW—upright walking; SW—stoop walking; KHC—knee and hand crawling, SKT: Skin temperature.

**Table 1 sensors-22-08094-t001:** Details of participants in the experiment.

Gender	Number	Age (Years)		Height (cm)		Weight (kg)		BMI (kg/m^2^)	
		Mean	SD	Mean	SD	Mean	SD	Mean	SD
Boys	13	5.0	0.82	116.31	6.84	22.07	4.48	16.17	1.71
Girls	15	5.4	0.91	116.20	5.85	20.75	2.93	15.34	1.42

Abbreviation: BMI, body mass index; SD, standard deviation.

**Table 2 sensors-22-08094-t002:** Average scores of physiological indicators.

Physiological Indicators		Baseline	UW	SW	KHC
		Mean	Mean	Mean	Mean
EDA	SC (μS)	10.26	19.92	20.31	22.14
	SCL (μS)	9.61	17.19	18.07	19.77
	SCR (μS)	0.65	2.82	2.24	2.45
SKT	(°C)	29	28.05	26.74	26.49

EDA: Electrodermal activity, SKT: Skin temperature, SC: Skin conductivity, SCL: Skin conductance level, SCR: Skin conductance response, UW—upright walking, SW—stoop walking, KHC—knee and hand crawling.

**Table 3 sensors-22-08094-t003:** Paired *t*-tests of the physiological measures between experiments.

Physiological Indicators		UW-Baseline	SW-Baseline	KHC-Baseline	UW-SW-KHC
		Sig.	Sig.	Sig.	Sig.
EDA	SC (μS)	0	0	0	0
	SCL (μS)	0	0	0.001	0
	SCR (μS)	0	0	0.18	0
SKT	(°C)	0	0	0	0

EDA: Electrodermal activity, SKT: Skin temperature, SC: Skin conductivity, SCL: Skin conductance level, SCR: Skin conductance response, UW—upright walking, SW—stoop walking, KHC—knee and hand crawling, Sig.: significance is *p* value.

**Table 4 sensors-22-08094-t004:** Baseline and KHC scores of physiological indicators for age.

Physiological Indicators	Baseline			KHC			Relative Change					
	4 Year	5 Year	6 Year	4 Year	5 Year	6 Year	Δ4	Δ5	Δ6	4 Year (%)	5 Year (%)	6 Year (%)
SC	10.90	14.82	9.28	23.38	30.87	17.34	16.05	12.48	8.06	114.50	108.30	86.77
SCL	10.09	13.88	8.64	21.50	27.02	14.90	13.14	11.41	6.26	113.08	94.67	72.45
SKT	30.68	30.42	30.75	28.55	28.37	28.78	−2.13	−2.05	−1.97	−6.94	−6.74	−6.41

KHC: Knee and hand crawling, SC: Skin conductivity, SCL: Skin conductance level, SKT: Skin temperature, Δ: KHC test value—baseline value.

**Table 5 sensors-22-08094-t005:** Baseline and KHC scores of physiological indicators for gender.

Physiological Indicators	Baseline (Mean)		KHC (Mean)			Relative Change		
	Boy	Girl	Boy	Girl	ΔBoy	ΔGirl	Boy (%)	Girl (%)
SC	13.63	9.71	28.95	18.78	15.32	9.07	112.40	93.41
SCL	12.63	9.11	26.21	16.07	13.58	6.96	107.52	76.40
SKT	31.15	26.61	27.84	25.92	−3.31	−0.69	−10.63	−2.59

KHC: Knee and hand crawling, SC: Skin conductivity, SCL: Skin conductance level, SKT: Skin temperature, Δ: posture test value − baseline value.

## Data Availability

The data that support the findings of this study are available upon reasonable request from the authors.

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
