# Peer review of "The Associations between Evacuation Movements and Children’s Physiological Demands Analyzed via Wearable-Based Sensors"

_sensors, 2022, doi:10.3390/s22218094_

Round 1
Reviewer 1 Report
Article Summary:
The article “The associations between evacuation movements and children's physiological demands analyzed via wearable-based sensors” is aimed at investigating the physiological demands in young children during emergency evacuation situations. It aims to inform the best practices for evacuation procedures for young children to minimize exposure to harsh environment. To that end, this article evaluates the velocity of children during different movement postures (upright walking, stooped walking and crawling).
Comments for Authors:
1. Authors must make their conclusions based on heights and BMI of children, in addition to age/gender?
2. It is unclear why age and gender were chosen as the basis for the comparison for different types of movement in children?
3. Why is EDA and SKT chosen for data collection? Does EDA include heart rate, respiratory rate (or other cardio respiratory parameters) to assess physiological impact? If so, authors should include that information in the manuscript. If not, authors should explain the reasoning behind choosing EDA, while avoiding use of heart and respiratory sensors.
4. Were wearable motion sensors considered for this experiment? Why were they not used to measure body motion and posture?
5. In Fig.8 and Fig.9, number of samples for each age is not mentioned.
6. What is the reason for all girls having the exact same KHC velocity? KHC velocity has a wider distribution amongst the boys. This should be discussed in the results section.
7. Authors claim the following in the ‘Discussions’ section [Lines 357-358]:
“Our new finding is that the evacuation velocities of children were faster than those of adults in UW and SW, however, the KHC velocity of children was slower than that of adults”
However, the manuscript does not include any measurements from adults, and therefore their claim is not supported by any results in this paper.
Authors must present their data and findings with more details (rather than single plots showing only mean or distribution of the test data).
Reviewer 2 Report
(Comment 1)
In this manuscript, the authors try to evaluate the effects of different locomotive postures on children’s velocity and physiological demands for effective evacuation by using wearable sensors.
The findings of this study will be contributed to improving the safety guide for children’s risk reduction.
From these contributions, I think that this paper is suitable for publication in Sensors. In order to publish, minor revisions with further several explanations might be necessary.
(Comment 2)
The introduction is clearly written. Especially, objectives and contributions are easy to understand. Good job.
(Comment 3)
Was trunk bending angle defined for the stoop walking (SW) condition?
(Comment 4)
The sampling/frame rate of wearable sensors and recorded video should be mentioned.
(Comment 5)
The resolution of Fig.5 , Fig.7, Fig.10, and Fig.11 should be improved.
(Comment 6)
Does floor condition such as surface or slope affect the velocity of each walk?
(Comment 7)
I am surprised by your finding that the physiological effects of postures are much greater than alarm sounds. I think this finding is very interesting. If you have several ideas for the reason of this result, please mention in the discussion.
(Comment 8)
Will you evaluate the mechanical loads or fatigue for bones or muscles of each walking posture? In my opinion, these loads and fatigue should be considered for evacuation over long distances.
(Comment 9)
The conclusion is clearly written. Good job.
Round 2
Reviewer 1 Report
The authors have provided additional information and addressed several of the previous concerns.
However, there remain a few questions that can greatly help improve the impact and quality of this research article:
1. The authors have added the Lines 268-271 where they speculate why the female subject population had a tighter KHC velocity distribution. However, their hypothesis regarding difference in body development and energy consumption is not supported by any data. It seems improbable that all 13 girls in this study have the exact same KHC velocity. The authors must show additional raw data plots and data processing algorithm as supplementary material to support their findings.
